# Efficient population size of urban agglomerations in the Yangtze River Economic Belt from a financial perspective

**Long Xiao** *

School of Media and Exhibition, Fujian Business University, Fuzhou, China

* xiaolongzi1991@fjbu.edu.cn

**Data Availability Statement:** All relevant data are within the manuscript.

**Funding:** This research was supported by the Major Project of the National social Science Foundation of China (20&ZD066).

## Abstract

Urban agglomerations, such as the Yangtze River Delta, Yangtze River Middle Reaches, and Chengdu-Chongqing regions, play a crucial role in driving China's regional economic development. While previous studies have focused on economic and social aspects, the fiscal dimension of urban agglomerations remains underexplored. This study addresses this gap by investigating the relationship between population size and fiscal efficiency in these three major urban agglomerations along the Yangtze River Economic Belt (YREB).We introduce the concept of fiscal efficiency based on revenue and expenditure and select relevant indices, such as efficient population size and fiscal self-reliance. Using statistical data from 2017 to 2019, we employ curve regression analysis in SPSS to estimate the efficient population sizes of these urban agglomerations and examine differences in financial efficiency over time and space. Our analysis reveals that cities with populations over 10 million hinder fiscal efficiency in the Yangtze River Delta, those with 3–5 and 5–10 million in the Yangtze River Middle Reaches, and those with 5–10 and 1–5 million in the Chengdu-Chongqing urban agglomerations. The maximum financially efficient population sizes are estimated at 648 million for the Yangtze River Delta, 308 million for the Yangtze River Middle Reaches, and 320 million for the Chengdu-Chongqing urban agglomerations. Considering various fiscal indicators, all three agglomerations demonstrate varying degrees of efficiency. The innovation of this study lies in the interdisciplinary approach, integrating finance, demography, urban planning, and regional economics. By analyzing population size from a fiscal perspective, we provide a novel theoretical framework and analytical tool for policymakers. This study highlights the importance of fiscal balance and population optimization in urban agglomerations, contributing to regional coordinated development and sustainable growth.

## 1. Introduction

Urban agglomeration represents the advanced stage of urban development, characterized by the clustering of cities into highly integrated systems. This phenomenon is not only prevalent in developed countries but also increasingly significant in rapidly developing nations like China. The Chinese government has identified urban agglomerations as pivotal to regional

**Competing interests:** The authors have declared that no competing interests exist.

development and has outlined strategies to foster their growth, particularly in key areas such as the Beijing-Tianjin-Hebei region, the Yangtze River Delta, the Guangdong-Hong Kong-Macao Greater Bay Area, the Chengdu-Chongqing region, the Yangtze River Middle Reaches, the Central Plains, and the Guanzhong Plain. Despite achieving an urbanization rate exceeding 60% by the end of 2020, there is a consensus that China's urbanization should ideally reach around 75%, considering the proportion of rural labor. The ongoing migration of populations from third- and fourth-tier cities to central urban areas is amplifying the effects of urban agglomeration, driving China's economic development. However, this rapid urbanization also brings challenges, including discrepancies in development needs and concepts across different regions. Against this backdrop, the study of urban agglomerations, particularly their effective population size, becomes crucial for understanding and addressing these issues.

Urban agglomeration is the highest form of spatial organization for cities that have entered the mature stage of development. In 2018, China issued the "Opinions of the Central Committee of the Communist Party of China and the State Council on Establishing a More Effective New Mechanism for Regional Coordinated Development." This document noted that urban agglomerations such as Beijing-Tianjin-Hebei, Yangtze River Delta, Guangdong-Hong Kong-Macao Greater Bay Area, Chengdu-Chongqing, Yangtze River Middle Reaches, Central Plains, and Guanzhong Plain can promote the integration and development of major regions and establish new models of development led by central cities. Although the urbanization rate of China's population had exceeded 60% by the end of 2020, some have suggested that the rate is still not where it should be. Calculated by the proportion of labor, the current proportion of rural labor is about 25%; thus, the urbanization level should reach 75%. Meanwhile, with the populations of third- and fourth-tier Chinese cities transferring to central urban areas, the effect of urban agglomeration is more pronounced, having become a driving force for China's future economic development. Yet, with the increased demand for rapid, high-quality urbanization, urban problems continue to emerge in the development of cities [1], and there are discrepancies in terms of urban development needs and concepts in areas such as suitability for business, livability, travel, and leisure. Thus, with urban agglomerations emerging as a recent form of urbanization, the size of urban agglomerations has become a topic of interest in urban research in China.

With the acceleration of urbanization, urban agglomerations have become an important force driving regional economic development. As an important growth pole of China's economy, the development of its urban agglomeration is particularly important in the Yangtze River Economic Belt. Therefore, studying the effective population size of the three major urban agglomerations on the Yangtze River Economic Belt is of great significance for promoting regional coordinated development and optimizing resource allocation. In previous studies on urban agglomerations, analysis was mostly conducted from economic, social, and other perspectives, while research from a fiscal perspective was relatively scarce. Finance, as an important means of government regulation of the economy, has a significant impact on the development of urban agglomerations. Therefore, studying the effective urban agglomeration population size of the Yangtze River Economic Belt from a financial perspective can help provide more comprehensive and scientific basis for government decision-making.

Based on city-size classifications in China, the sizes of urban agglomerations can also be subdivided according to, for example, population size, economic level, land area, and comprehensive size. This study mainly focuses on the population size of urban agglomerations. Existing research focuses on areas such as the spatial characteristics of urban agglomeration populations, as well as the related distribution and evolution trends, economic performance, spatial benefits, and city resilience patterns. As early as the mid-twentieth century, Western researchers such as noted that the distribution of urban size follows some general law. These

general laws discovered by Western scholars also exist in the development process of urban agglomerations in China. For instance, by calculating the primacy degree, regression slope, and city size Gini index of the Pearl River Delta, Yu et al. [2] uncovered the basic law that the urban population size of the Pearl River Delta urban agglomeration tends to be dispersed, and its population size shows spatial characteristics such as fan agglomeration, spatial directivity, growth axiality, and regional differences. Chen et al. [3], meanwhile, studying the Yangtze River Delta metropolitan area, found that it urban population distribution tended to be scattered, and its first-tier cities held a strong monopoly position while the middle-tier cities had lower development levels and smaller populations. Wang and Gao [4] studied the effect of the spatial structure of the population on economic performance in the Yangtze River Middle Reaches urban agglomeration; they found that the population's multicenter distribution significantly promoted economic performance, and population size was a key factor affecting economic performance. In multiple studies, Wang et al. [5] studied population size distribution and evolution in the urban agglomerations of Beijing-Tianjin-Hebei, the Yangtze River Middle Reaches, Chengdu-Chongqing, and the Central Plains. They found that Beijing-Tianjin-Hebei showed divergent growth trends, and Beijing's agglomeration effect was predominant while small and medium-sized cities lagged in their development. Furthermore, overall urbanization in the Yangtze River Middle Reaches agglomeration was not high, the leading city was not strong, and the urban system showed a trend of agglomeration first, followed by decentralization [6]. Chengdu-Chongqing showed a "dual-core" type of population distribution; the agglomeration effect of its big cities was gradually increasing, the urbanization level was relatively large, and its industrial structures were relatively similar [7]. Finally, the Central Plains showed an olive-shaped urban population size structure, whose distribution had a fault phenomenon; furthermore, the development of small cities was insufficient, and they were relatively small in size with small populations [8]. Zhang and Xiong [9] noted that the city population size distribution of the Yangtze River Middle Reaches urban agglomeration had undergone adjustments in terms of centralization and equilibrium since 2001, with an overall trend toward equilibrium. Zhong et al. [10] investigated the spatial structure of the Yangtze River Middle Reaches urban agglomeration in 2017 in terms of population size benefits, distance benefits, and spatial connection benefits; they found that the Chang-Zhu-Tan agglomeration had advantages in terms of population and economic size. Lastly, Chen and Xia [11] analyzed the spatiotemporal relationships between population size and urban resilience in the Yangtze River Middle Reaches urban agglomeration; they found that the larger the urban population, the higher the urban resilience.

Although many studies have investigated population size in China's urban agglomerations, most are limited to a few agglomerations (e.g., Beijing-Tianjin-Hebei, Yangtze River Middle Reaches, Pearl River Delta, and Yangtze River Delta), with little research on other national-level urban agglomerations. Meanwhile, such studies mostly rely on commonly used indicators and are not innovative. In addition, many studies replace the population size of urban agglomerations with the size of urban populations, or the research topic is urban agglomeration population size, but they settle on urban population size, which cannot reflect the overall situation of urban agglomerations. This study, therefore, examines the overall efficient population size of three urban agglomerations in the Yangtze River Economic Belt (YREB): the Yangtze River Delta, Yangtze River Middle Reaches, and Chengdu-Chongqing urban agglomerations. In summary, existing studies generally acknowledges the significant interplay between the population size of urban agglomerations and economic development, as well as fiscal efficiency. On one hand, the scale effect brought about by population agglomeration facilitates the concentration of economic activities, enhances resource allocation efficiency, and consequently strengthens fiscal revenue capabilities. On the other hand, rational investment and allocation

of fiscal funds can further propel infrastructure construction, public service improvements, and industrial structure optimization, providing robust support for the expansion of population size and the enhancement of population quality. Nevertheless, existing research also points out several issues pertaining to population size and fiscal efficiency within urban agglomerations. For instance, excessive population concentration in some cities leads to "urban diseases" such as traffic congestion and environmental pollution, which increase fiscal burdens and reduce fiscal efficiency. Meanwhile, during the process of population expansion, some cities have neglected the rational use of fiscal funds, resulting in lagging infrastructure and public service construction, which hinders the improvement of population quality and sustainable urban development. In response to these issues, existing research has proposed various solutions. Firstly, regional coordination should be strengthened to optimize the spatial distribution of population, achieving efficient resource allocation and complementary advantages through staggered development within urban agglomerations. Secondly, the efficiency of fiscal fund utilization should be improved, with enhanced budget management and performance evaluation to ensure that fiscal funds are genuinely utilized to promote economic development and social progress.

Based on the current research status mentioned above, this study investigates the effective population size of the three major urban agglomerations on the Yangtze River Economic Belt from a financial perspective, filling the gap in previous research and possessing unique innovation. On the one hand, it helps to enrich and improve the theoretical system of urban agglomerations, and promote the development of related disciplines. At the same time, the research results can also provide reference and inspiration for the study of urban agglomerations in other regions. On the other hand, research findings can provide practical guidance for the urban agglomeration development of the Yangtze River Economic Belt and promote sustainable regional economic development.

The research process of this paper is shown in Fig 1. This study focuses on the effective population size of the three major urban agglomerations along the Yangtze River Economic Belt (YREB): the Yangtze River Delta, the Yangtze River Middle Reaches, and the Chengdu-Chongqing urban agglomerations. We adopt a fiscal perspective, recognizing the significant role finance plays in shaping urban agglomerations and influencing government decisions. Previous studies have primarily examined urban agglomerations from economic and social viewpoints, with less attention paid to the fiscal implications. To address this gap, we will analyze the relationship between population size and fiscal efficiency, aiming to provide a more comprehensive understanding for policymakers. The methodology involves a detailed review of existing literature, quantitative analysis of population and fiscal data, and comparative case studies. The remainder of this paper is organized as follows: the Methods section provides a detailed literature review, the Results section outlines the methodology and data sources, the Discussion section presents the results and discussion, and the Conclusion section concludes with policy implications and suggestions for future research.

## 2. Methods

### 2.1. Research area

As of February 18, 2019, China's State Council had approved 10 national-level urban agglomerations: Yangtze River Middle Reaches, Harbin-Changchun, Chengdu-Chongqing, Yangtze River Delta, Central Plains, Beibu Gulf, Guanzhong Plain, Hubao Eyu, Lanxi, and Guangdong-Hong Kong-Macao Greater Bay Area. Among them, this study focused on the efficient population sizes of three, all located in the YREB: Yangtze River Delta, Yangtze River Middle

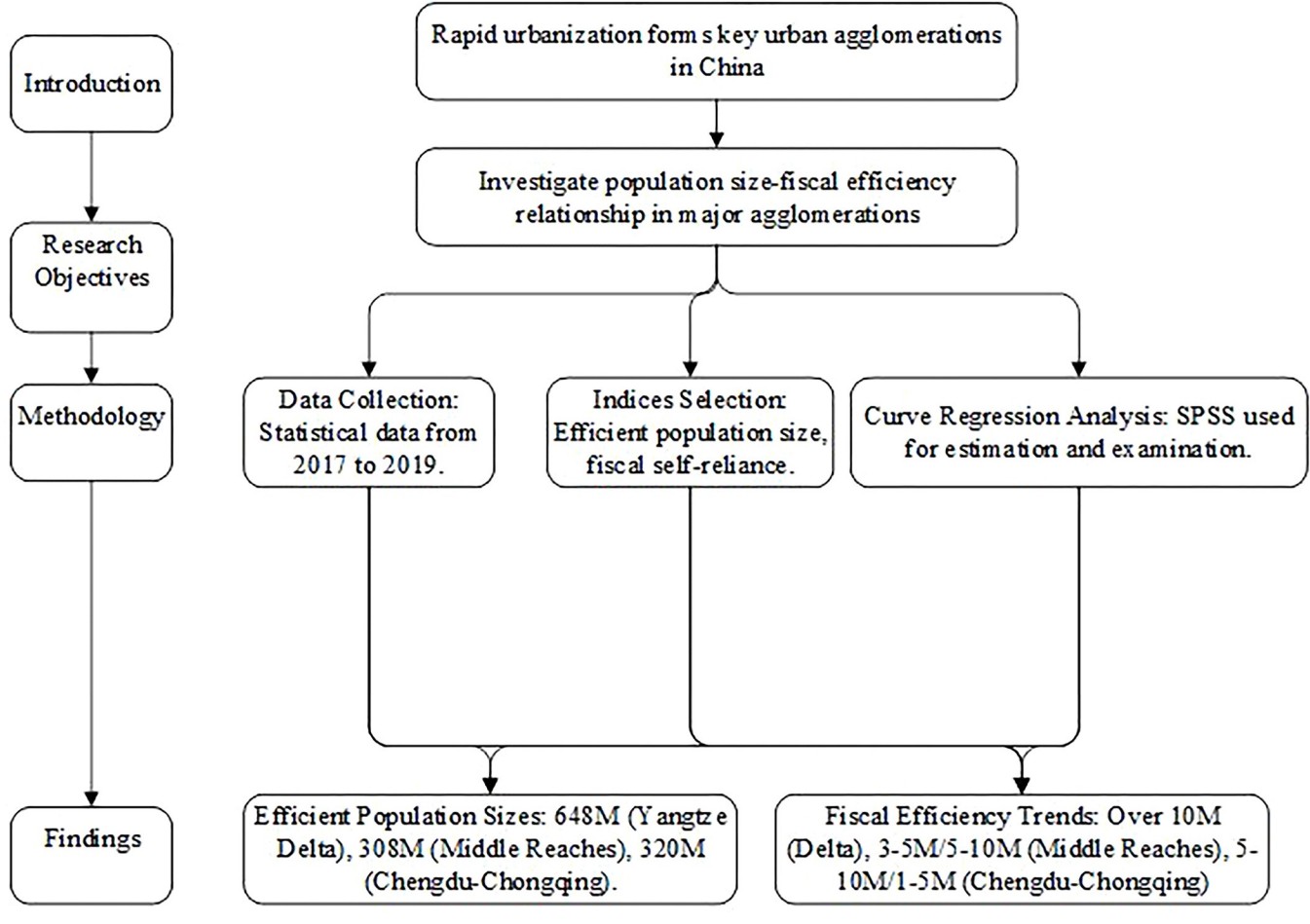

**Fig 1. Research process.**

Reaches, and Chengdu-Chongqing (Fig 2 is originally drawn by the author, there are no copyright disputes).

China's State Council issued the "Approval of the State Council on the Development Plan of Urban Agglomeration in the Middle Reaches of the Yangtze River" in 2015. At the end of 2019, the resident population of this city cluster was about 130 million people. Its land area was 317,000 square kilometers and included the following cities: Wuhan, Huangshi, Ezhou, Huanggang, Xiaogan, Xianning, Xiantao, Qianjiang, Tianmen, Xiangyang, and Yichang in Hubei Province; Jingzhou, Jingmen, Changsha, Zhuzhou, Xiangtan, Yueyang, Yiyang, Changde, Hengyang, and Loudi in Hunan Province; and Nanchang, Jiujiang, Jingdezhen, Yingtan, Xinyu, Yichun, Pingxiang, Shangrao, and parts of Fuzhou and Ji'an in Jiangxi Province. Meanwhile, the "Approval of the State Council on the Development Plan of the Chengdu-Chongqing Urban Agglomeration" was issued in 2016. In 2019, that agglomeration had 100 million people and a land area of 185,000 square kilometers. It included most of Chongqing City, Chengdu, Zigong, Luzhou, Deyang, Suining, Neijiang, Leshan, Nanchong, Meishan, Yibin, Guang'an, Ziyang, and parts of Mianyang, Dazhou, and Ya'an in Sichuan Province. Lastly, the "Approval of the State Council on the Development Plan of the Yangtze River Delta Urban Agglomeration" appeared in 2016. At the end of 2019, its population was 163 million people, its land area was 358,000 square kilometers, and its central area was

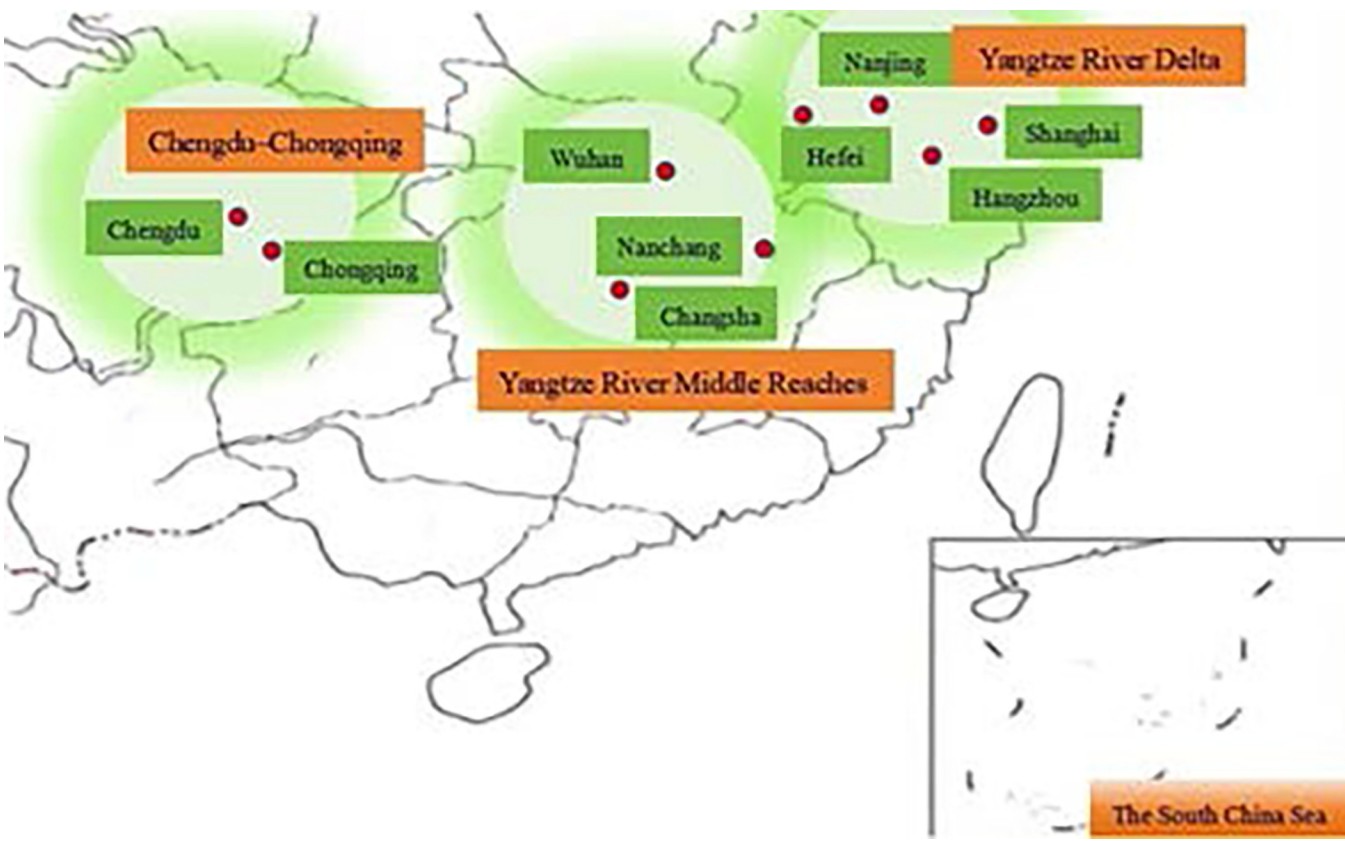

**Fig 2. Overview map of the study area: Yangtze River Delta, Yangtze River Middle Reaches, and Chengdu-Chongqing (orange).**

225,000 square kilometers. The agglomeration includes Shanghai, Nanjing, Wuxi, Changzhou, Suzhou, and Nantong in Jiangsu Province; Yancheng, Yangzhou, Zhenjiang, Taizhou, Hangzhou, Ningbo, Jiaxing, Huzhou, Wenzhou, Shaoxing, Jinhua, Zhoushan, and Taizhou in Zhejiang Province; and Hefei, Wuhu, Ma'anshan, Tongling, Anqing, Chuzhou, Chizhou, and Xuancheng in Anhui Province.Specific data sources are shown in Table 1.

Our research area includes all of the Yangtze River Delta and Chengdu-Chongqing urban agglomerations and most of the Yangtze River Middle Reaches urban agglomeration. The statistical scope is consistent with official divisions, including both urban and rural areas under the jurisdictions of cities. Because of the unavailability and non-stripability of some data, this study eliminated the three county-level cities of Xiantao, Qianjiang, and Tianmen in the Yangtze River Middle Reaches urban agglomeration, which are directly under the central government of Hubei Province. This exclusion does not affect the final results. Furthermore, all of the city of Ji'an is included in the Yangtze River Middle Reaches urban agglomeration. Finally, the Chengdu-Chongqing agglomeration also includes the entire territory data for Chongqing, Mianyang, Dazhou, and Ya'an.

## 2.2. Data sources

The data required for this study involve the populations of urban agglomerations, fiscal revenues, and fiscal expenditures. Considering data availability and applicability, this study use permanent population to measure the population of urban agglomerations. Urban agglomerations are used in accordance with the common measurement methods of fiscal

**Table 1. Data sources.**

| Urban Agglomeration | Year of Approval | Resident Population (2019) | Land Area (sq km) | Cities Included |
|---|---|---|---|---|
| Yangtze River Middle Reaches | 2015 | 130 million | 317,000 | Hubei: Wuhan, Huangshi, Ezhou, Huanggang, Xiaogan, Xianning, Xiantao, Qianjiang, Tianmen, Xiangyang, Yichang<br>Hunan: Changsha, Zhuzhou, Xiangtan, Yueyang, Yiyang, Changde, Hengyang, Loudi<br>Jiangxi: Nanchang, Jiujiang, Jingdezhen, Yingtan, Xinyu, Yichun, Pingxiang, Shangrao, parts of Fuzhou and Ji'an |
| Chengdu-Chongqing | 2016 | 100 million | 185,000 | Chongqing: Most of the city<br>Sichuan: Chengdu, Zigong, Luzhou, Deyang, Suining, Neijiang, Leshan, Nanchong, Meishan, Yibin, Guang'an, Ziyang, parts of Mianyang, Dazhou, and Ya'an |
| Yangtze River Delta | 2016 | 163 million | 358,000 (central area: 225,000 sq km) | Shanghai: Entire city<br>Jiangsu: Nanjing, Wuxi, Changzhou, Suzhou, Nantong, Yancheng, Yangzhou, Zhenjiang, Taizhou<br>Zhejiang: Hangzhou, Ningbo, Jiaxing, Huzhou, Wenzhou, Shaoxing, Jinhua, Zhoushan, Taizhou<br>Anhui: Hefei, Wuhu, Ma'anshan, Tongling, Anqing, Chuzhou, Chizhou, Xuancheng |

revenue and expenditure. The sum of local general public budget revenues and the sum of local general public budget expenditures in each city replace urban agglomeration fiscal revenue and urban agglomeration fiscal expenditures, respectively. The data mainly come from the "China Statistical Yearbook," "Statistical Bulletin," "Statistical Yearbooks," and "China City Statistical Yearbooks" of various provinces and cities. When individual data are missing, this study use the average method for supplementary processing based on the collected data.

## 2.3. Research index

**2.3.1. Efficient population size.** As populations shift toward urban agglomerations, the agglomeration effect continues to increase, and the fiscal revenues and expenditures of agglomerations continue to rise as well. On the one hand, increases in population size will promote fiscal revenue generation and become a source of agglomerations' finances. On the other hand, population concentration further diversifies public needs, and meeting those needs expands financial expenditures in the operating expenses of urban agglomerations. In other words, the fiscal revenues and expenditures of urban agglomerations are functions of population. Therefore, from the perspective of cost-effectiveness, when a reasonable ratio between fiscal revenue and fiscal expenditure is achieved, the size of the urban agglomeration will be optimal. This ratio is the efficiency of the urban agglomeration, and the population size corresponding to efficiency is the efficient population size [12].

Based on Nakamura and Kanauchi's concept of efficient population size, this study construct a theoretical model of an urban agglomeration's efficient population size [12]:

$$\begin{cases} Y = f_1(N) \\ C = f_2(N) \end{cases} \qquad (1)$$

In Eq (1), Y is fiscal revenue, C is fiscal expenditure, and N is population. Then, aiming to seek a reasonable ratio between income and expenditure, this study construct an efficiency function.

In the following formula, E is the efficiency index:

$$E = \frac{Y}{C} = \frac{f_1(N)}{f_2(N)}$$

(2)

By deriving N in Eq (2), the maximum value of Y/C can be calculated. When Y/C takes the maximum value, the corresponding population is the efficient population. If the maximum value does not exist, the efficient population can be determined through reasonable consideration.

The first Eq (1) indicates that the efficiency index E is the ratio of the output value Y to the input value C. This ratio can be used to measure the efficiency of how much output a system or process produces for a given input. The second Eq (2) is a more concrete expression, where E denotes the efficiency index, Y/C denotes the ratio of output to input, $f_1$ (N) and $f_2$ (N) denote two functions respectively, which depend on the variable N. By solving Eq (2), one can calculate the value of N at which Y/C reaches a maximum. This value of N is called the "efficient population" and represents the number of people needed to achieve the maximum output/input ratio in the optimal case.

If a maximum does not exist, an efficient population can be determined by reasonable consideration.

**2.3.2. Fiscal independence.** It is desirable for city clusters to have a surplus in their finances. However, to cope with the size of the efficient population under a budget deficit, this study propose a fiscal independence index based on Mori [13]. Fiscal independence is the ratio of deficit to fiscal revenue; within 10% is generally considered best. When the degree of fiscal independence is within 10%, the corresponding population size is efficient [14].

$$I = \frac{D}{Y}$$

(3)

In Eq (3), I is the fiscal independence index, and D is the deficit.

# 3. Results

## 3.1. Fiscal revenue and expenditure-population model of urban agglomerations in the YREB

In Fig 3, the horizontal axis is the natural logarithm of the population (million people) of each city in the urban agglomeration; the vertical axis is the natural logarithm of the city's fiscal revenue (100 million yuan) and fiscal expenditure (100 million yuan). Furthermore, using regression in SPSS, this study create an approximate trend curve of the income and expenditure-population relationship. Any point on the trend curve is the average value of the city in the urban agglomeration (income/expense average, population average), and the average line can reflect the overall trend of these indicators of the urban agglomeration [14]. Based on preliminary attempts, the curve estimation in the regression can simulate a better curve than other methods, and multiple submethods in curve estimation can achieve the simulation of the curve. This study select the most suitable submethod based on our research needs. Fig 3 shows the results.

In Fig 3, display a statistically significant trend curve model of urban agglomeration fiscal revenue and fiscal expenditure. It is simulated by curve estimation based on member city data as the population increases. In the above models, the corresponding t-values of the explanatory variables are all above 1.65, which means they pass the significance test at the 10% level, and the coefficient of determination also indicates that the model has explanatory power.

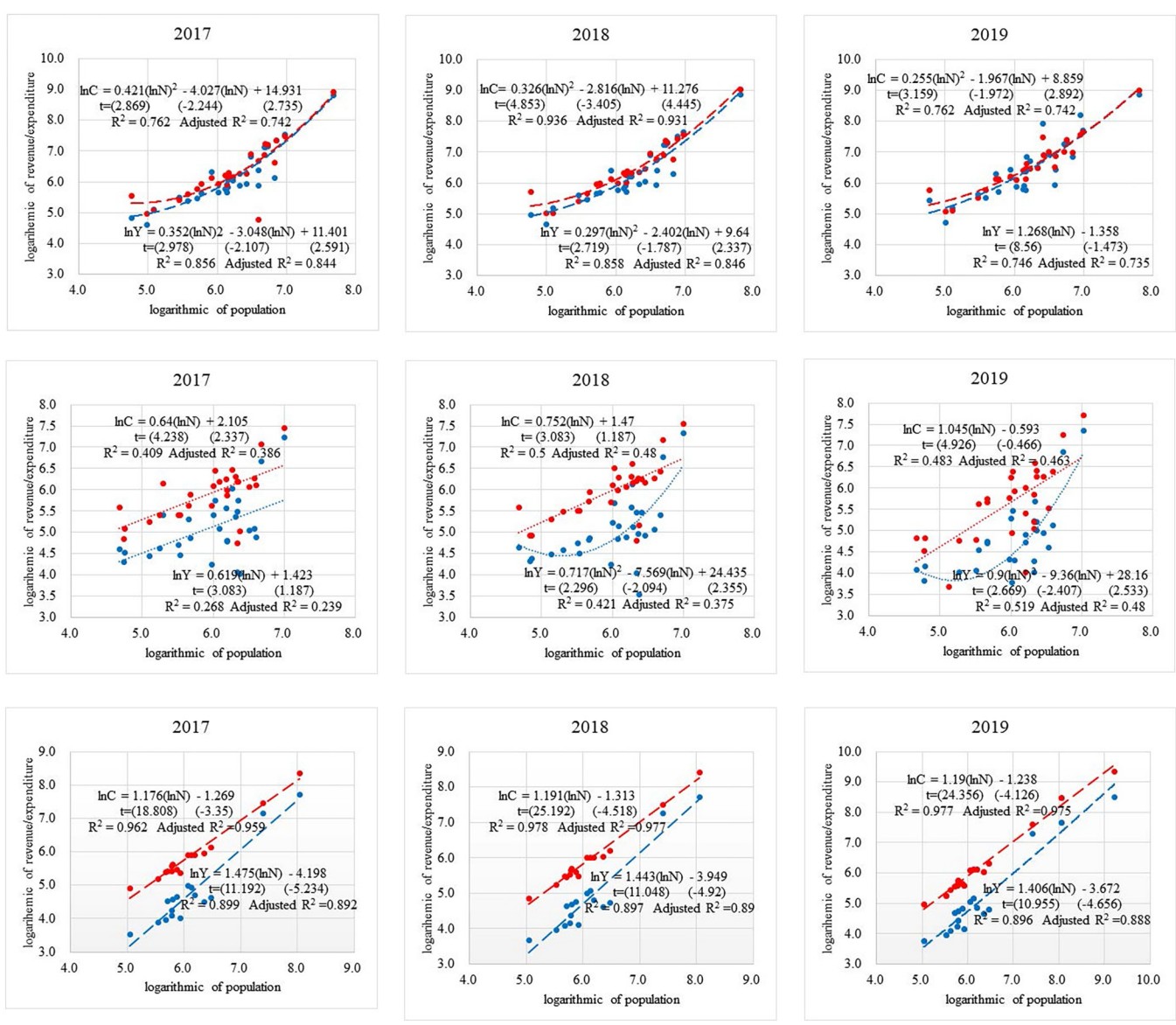

**Fig 3. Fiscal revenue (blue), fiscal expenditure (red), and population size of urban agglomerations.** (a) Yangtze River Delta urban agglomeration. (b) Yangtze River Delta urban agglomeration. (c) Yangtze River Delta urban agglomeration. (d) Yangtze River Middle Reaches urban agglomeration. (e) Yangtze River Middle Reaches urban agglomeration. (f) Yangtze River Middle Reaches urban agglomeration. (g) Chengdu-Chongqing urban agglomeration. (h) Chengdu-Chongqing urban agglomeration. (i) Chengdu-Chongqing urban agglomeration.

In Fig 3, the trend curves of fiscal revenue and expenditure with respect to the populations of the three selected urban agglomerations extend upward to the right with increasing population (any point on the trend curve is a one-to-one correspondence between the average urban population and average urban income/expenditure). The three-year fiscal revenue trend curve of the Yangtze River Delta urban agglomeration shows a quadratic curve; the fiscal expenditure trend curve also shows a quadratic curve, except for a linear straight line in 2019. With continuously increasing population, fiscal revenue constantly approaches fiscal expenditure and eventually exceeds the trend of fiscal expenditure. The three-year fiscal revenue trend curve of the Yangtze River Middle Reaches urban agglomeration is a quadratic curve, except for a linear straight line in 2017, and the expenditure trend curve is a linear straight line. Revenue and

expenditure in 2017 expand with population size. As the population expands in 2018 and 2019, income first deviates from the expenditure line and reaches the lowest level; it then continues to move closer to the expenditure line and finally closes to form a smiling curve. The fiscal revenue and expenditure trend curves of the Chengdu-Chongqing urban agglomeration are all linear. With continuous population increases, the income line continues to approach the expenditure line, and the gap between income and expenditure shrinks over time.

Based on the idea of "$\ln\left(\frac{Y}{C}\right) = \ln(Y) - \ln(C)$" with the help of the equations in the above model, this study can deduce the specific fiscal efficiency of urban agglomerations as a function of population.

## 3.2. Financial efficiency-population model and analysis of urban agglomerations in the YREB

Using the financial revenue and expenditure equations in the previous section, and based on the formula $E = Y/C = e^{(lnY - lnC)}$, this study can see that the financial efficiency of urban agglomeration is a function of urban agglomeration population. Then, continuing to use the urban agglomeration data for financial revenue, financial expenditure, and population, this study use SPSS to simulate the functional model and curve of financial efficiency E and population N of the urban agglomerations (Fig 4).

Any point on the simulated curve in Fig 4 represents financial efficiency corresponding to the theoretical average urban population. The population size of an urban agglomeration is equal to the average urban population multiplied by the number of member cities. The trend curve can therefore reflect the relationship between the overall efficiency of an urban agglomeration and the population. In Fig 4, the financial efficiency-population relationship in the three investigated urban agglomerations shows three different trends. Our findings for the Yangtze River Delta and Yangtze River Middle Reaches urban agglomerations are similar to the effective urban population size curves obtained by Xiao et al. [15] based on the income cost curve. Specifically, the two agglomerations have not yet reached the maximally efficient population size, and there is still potential for more population growth. The population of the Chengdu-Chongqing urban agglomeration is consistent with the inverted U-shaped effective population size curve identified by Camagni et al. [16] and Mori [13].

The financial efficiency of the Yangtze River Delta urban agglomeration shows a power curve trend of rising with population size. According to the three-year results, the financial efficiency and average urban population size of the Yangtze River Delta urban agglomeration have not reached the highest level. Average financial efficiency in the agglomeration is about 0.9, and the maximum efficiency is about 1.1. However, taking the average urban population size of 10 million people (the corresponding financial efficiency value is about 1.0) as the watershed, financial efficiency rises rapidly when the urban population is less than 10 million, and the improvement of financial efficiency becomes slower with the gradual increase in population when the urban population is more than 10 million. Specifically, the financial efficiency of the Yangtze River Delta urban agglomeration is more evenly distributed among cities with different population sizes. Among cities with a population of more than 10 million, Shanghai and Suzhou have average financial efficiency in the agglomeration while Hangzhou has the highest financial efficiency. The financial efficiency of cities with populations of 5–10 and 1–5 million in the Yangtze River Delta urban agglomeration is evenly distributed around the trend curve, with an efficiency value of about 0.9. Regarding the relationship between financial efficiency and population size, the Yangtze River Delta urban agglomeration has achieved coordinated development and complementary advantages and disadvantages among cities with various population sizes. If the financial efficiency of Shanghai, Suzhou, and other megacities

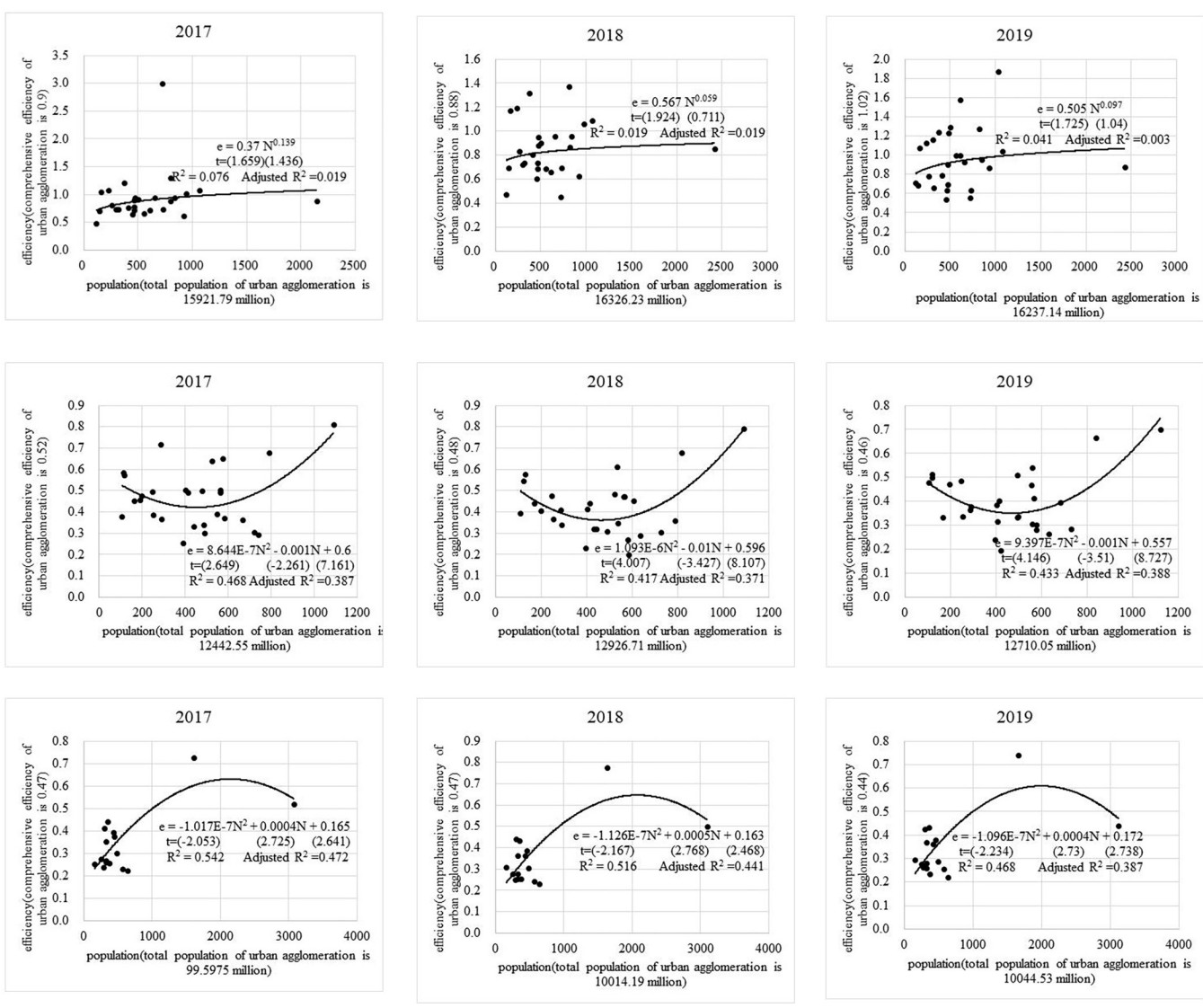

**Fig 4. Urban agglomeration efficiency and population size.** (a) Yangtze River Delta urban agglomeration. (b) Yangtze River Delta urban agglomeration. (c) Yangtze River Delta urban agglomeration. (d) Yangtze River Middle Reaches urban agglomeration. (e) Yangtze River Middle Reaches urban agglomeration. (f) Yangtze River Middle Reaches urban agglomeration. (g) Chengdu-Chongqing urban agglomeration. (h) Chengdu-Chongqing urban agglomeration. (i) Chengdu-Chongqing urban agglomeration.

can be further improved in the future, this study can expect that with continuous increases in the population size of the entire urban agglomeration, the overall financial efficiency of the Yangtze River Delta urban agglomeration will still be significantly improved.

The trend curve of the relationship between financial efficiency and population in the Yangtze River Middle Reaches urban agglomeration shows a U-shaped trend of first decreasing and then increasing with increasing population. Average financial efficiency in the Yangtze River Middle Reaches urban agglomeration is about 0.5, and the corresponding maximum average urban population size is about 8 million. Specifically, the financial efficiency of cities with populations of 1–3 and more than 8 million is relatively high, while cities with relatively low financial efficiency are mostly those with populations of 3–8 million. The main reason for the trend curve is the low financial efficiency of cities with populations of 3–8 million, which

also lowers the overall financial efficiency of the agglomeration. Although the financial efficiency of this urban agglomeration shows some trends of rising with population, in terms of the national strategy of coordinating the development of cities of all sizes and types, the spatial correlation within the Yangtze River Middle Reaches urban agglomeration cannot be ignored [17]. This urban agglomeration still needs to find ways to improve the financial efficiency of most cities with populations of 3–8 million, since those cities are the main driving force for the sustainable expansion of efficient population size in the future.

The financial efficiency-population trend curve of the Chengdu-Chongqing urban agglomeration is in line with the inverted U-shaped curve dominated by academia, and the average financial efficiency value is about 0.45. In Fig 4, the highest financial efficiency value of the Chengdu-Chongqing urban agglomeration is about 0.6, and it will reach the average level with an urban population of 20 million. Overall, the financial efficiency-population relationship curve of this urban agglomeration reflects its dual-center development—namely, Chongqing and Chengdu account for the highest financial efficiency and highest average urban population sizes in the agglomeration. The financial efficiency of cities with populations of 1–10 million in the agglomeration is low. The biggest disadvantage of this agglomeration's "dual-center" model is the lack of coordinated development within the urban agglomeration, which is not conducive to balanced regional development. Thus, the key to healthy, sustainable, coordinated development in the Chengdu-Chongqing urban agglomeration lies in cities with populations of 1–10 million.

Taking the current highest financial efficiency as a reference, the average urban population size of the Yangtze River Delta, Yangtze River Middle Reaches, and Chengdu-Chongqing urban agglomerations can theoretically reach 24, 11, and 20 million, respectively, and the total population size of the urban agglomeration can reach 648, 308, and 320 million, respectively. The population with the highest financial efficiency among the three urban agglomerations is about 2.5–4.0 times the current real resident population.

## 3.3. Efficient population size of urban agglomerations in the YREB under fiscal independence

Here, this study introduce the financial independence index (ratio of the absolute value of financial deficit to financial revenue) and consider the financial independence of urban agglomerations based on financial independence, population, financial revenue, financial expenditure, fiscal revenue-expenditure gap, and other relevant data and indicators. Then, this study analyze the financial efficiency and efficient populations of the urban agglomerations. For 2017–2019, the three agglomerations mostly showed fiscal deficits. The fiscal deficits of the Yangtze River Delta urban agglomeration in 2017 and 2018 were 228.1 and 337.5 billion yuan, respectively; fiscal surplus began in 2019 (53.7 billion yuan). The three-year fiscal deficits of the Yangtze River Middle Reaches and Chengdu-Chongqing urban agglomerations fluctuated within a range of 600–700 and 700–800 billion yuan, respectively. The fiscal deficits of the Yangtze River Middle Reaches urban agglomeration have decreased.

If the absolute value of the financial independence index is less than 10% in the case of fiscal surplus or deficit, the urban agglomeration has good self-reliance ability, and the corresponding population size is efficient Mori [13]. Based on the data in the last two rows in Table 2, this study can see that the financial independence of the three urban agglomerations is lower than that of the provinces and cities where they are located. That is, if the three urban agglomerations in the YREB are independent, their deficit level will be reduced, which is conducive to improved development. Among them, the Yangtze River Middle Reaches and Chengdu-Chongqing urban agglomerations have great difficulty with financial self-reliance, which

**Table 2. Results of indicators related to the self-reliance of urban agglomeration.**

| Indicator | Year | Yangtze River Delta urban agglomeration | Middle Yangtze River urban agglomeration | Chengdu-Chongqing urban agglomeration |
|---|---|---|---|---|
| Proportion of urban agglomeration population in the total population of provinces and cities (%) | 2017 | 72.08 | 76.50 | 87.54 |
| | 2018 | 72.45 | 76.58 | 87.52 |
| | 2019 | 72.26 | 76.81 | 87.35 |
| Proportion of general public budget revenue of urban agglomeration in total general public budget revenue of provinces and cities (%) | 2017 | 87.60 | 86.38 | 81.29 |
| | 2018 | 87.49 | 87.37 | 81.43 |
| | 2019 | 89.81 | 88.43 | 81.27 |
| Proportion of general public budget expenditure of urban agglomeration in total general public budget expenditure of provinces and cities (%) | 2017 | 81.59 | 78.80 | 77.16 |
| | 2018 | 82.41 | 78.83 | 74.95 |
| | 2019 | 82.27 | 81.10 | 80.97 |
| Proportion of fiscal revenue and expenditure difference of urban agglomeration in the total fiscal revenue and expenditure difference of provinces and cities (%) | 2017 | 49.50 | 72.02 | 73.81 |
| | 2018 | 58.56 | 72.37 | 70.00 |
| | 2019 | 22.51 | 75.77 | 80.73 |
| Financial self-reliance of provinces and cities in the urban agglomeration (%) | 2017 | 18.73 | 111.76 | 123.28 |
| | 2018 | 21.27 | 132.26 | 130.87 |
| | 2019 | 7.20 | 137.66 | 128.25 |
| Financial self-reliance of the urban agglomeration (%) | 2017 | 10.58 | 93.18 | 111.94 |
| | 2018 | 14.24 | 109.55 | 112.49 |
| | 2019 | Surplus | 118.96 | 127.40 |

means these two agglomerations are still financially dependent on their provinces/cities and the central government. The Yangtze River Delta urban agglomeration has good financial self-reliance, and its population size is efficient.

According to all indicators in Table 2, the Yangtze River Delta urban agglomeration has created 87% or more of its fiscal revenue, relying on 72% of the permanent residents in the four provinces and cities where it is located. Its fiscal expenditure only accounted for about 82% of expenditures in the four provinces and cities. This study can see that the Yangtze River Delta urban agglomeration has created more fiscal revenue with its own population; this can not only meet its own needs but also provide feedback to local provinces/cities and the central government. In addition, in combination with the ratio of urban agglomeration to provincial and municipal revenue and the expenditure gap, the ratio of the fiscal revenue-expenditure gap of the three agglomerations to the provincial and municipal financial revenue-expenditure gap is lower than the corresponding population, income, and expenditure ratios. From the financial perspective, this study can see that the three urban agglomerations in the YREB still have good development momentum, but it will take time to improve the status quo.

In conclusion, in terms of fiscal self-reliance, only the Yangtze River Delta urban agglomeration has good efficiency, and its corresponding population size is efficient, if this study follow Mori's [13] approach. Based on the analysis of all indicators in Table 1, the Yangtze River Middle Reaches and Chengdu-Chongqing urban agglomerations have financial deficits, but their financial independence is not below the standard of 10%. Considering that China's urban agglomerations receive financial support from their provinces/cities and the central government, and they still have some degree of self-reliance and the financial capacity to feed back to their provinces and cities, it is quite possible for them to overcome their financial difficulties in the future. Considering all indicators as well as further reasonable population distribution and adjustment within the urban agglomerations, if the coordinated development of multiple types

of cities can be achieved [18], the population sizes of these two urban agglomerations will still be efficient.

## 4. Discussion

At present, research on urban agglomerations mostly stays at the level of review and qualitative description, with less empirical research and a lack of comparative analysis of detailed spatio-temporal data. Existing research mainly focuses on urban agglomerations in the northeast of the United States, Pacific coastal urban agglomerations in Japan, London urban agglomerations in the United Kingdom, and northwestern European urban agglomerations. For China's urban agglomerations, which are still undergoing industrial transformation, the trend of population aggregation within urban agglomerations towards central cities continues. However, the large outflow of population due to insufficient economic development vitality has become an important factor affecting regional coordinated development, and serious problems such as the lack of effective coordination between central cities and other small and medium-sized cities in urban agglomerations have also emerged. From the perspective of urban agglomeration development, there are still a few mature urban agglomerations among the 19 major urban agglomerations in China. Most urban agglomerations are undergoing a critical stage of transformation, and population and financial development have not yet formed a good collaborative relationship, such as the Harbin Great Wall urban agglomeration, the Liaozhong South urban agglomeration, the Yangtze River middle reaches urban agglomeration, and the Chengdu Chongqing urban agglomeration [19]. Among the three urban agglomerations discussed in this study, only the Yangtze River Delta urban agglomeration has formed a good cooperation relationship between population and finance.

Zipf [20] pioneered the introduction of the law of universal gravitation in physics to analyze the spatial structure of urban agglomerations. Subsequently, empirical research on the spatial structure of urban agglomerations flourished, and scholars studied the spatial structure of urban agglomerations from multiple perspectives. Friedmann [21] simulated the stages and processes of spatial structure evolution in urban agglomerations based on the "Core-Edge" theory. Yu [22] studied the spatial structure evolution process and characteristics of the Beijing metropolitan area by constructing regional space. Chin [23] constructed a statistical model on population employment and commuting data to empirically study the spatial structure of metropolitan areas in the Midwest of the United States. Tsai [24], Meijers and Burger [25] describe spatial structure from three aspects: scale dimension, single center—multi center dimension, and agglomeration—diffusion dimension. The innovation in methodology of this study is mainly reflected in the introduction of a fiscal perspective, the combination analysis of efficiency and population size, multidimensional data analysis, innovative model construction, and the combination of dynamic and static analysis. The use of these innovative methods can more comprehensively reveal the relationship between population size and fiscal efficiency in the national level urban agglomeration of the Yangtze River Economic Belt, and make up for the shortcomings of existing research from a new perspective.

Discussion on the stages of population spatial distribution in urban agglomerations. The "Spatial Cycle Hypothesis" suggests that the population spatial structure of urban agglomerations will undergo processes of urbanization, suburbanization, reverse urbanization, and re urbanization [26]. There are also studies suggesting that the population spatial distribution of urban agglomerations will undergo three stages: linear decline, inverted U-shaped distribution, and "M-shaped" distribution [27]. This study found that the population spatial distribution of the Yangtze River Delta urban agglomeration, the middle reaches of the Yangtze River urban agglomeration, and the Chengdu Chongqing urban agglomeration showed different,

such as exponential increase, U-shaped distribution, and inverted U-shaped distribution, respectively. And filling the gap in existing research on population spatial distribution in urban agglomerations.

Recent studies have explored various aspects of socio-economic development and environmental impacts in the Yangtze River Economic Belt (YREB). Cai and Tu [28] analyzed the spatiotemporal characteristics and driving forces behind construction land expansion in the YREB, highlighting the complex dynamics shaping urban growth. Chen and Shen [29] examined the spatiotemporal differentiation of urban-rural income disparity and its driving forces between 2000 and 2017, providing insights into regional inequality. Li et al. [30] investigated the spatial spillover effects of urban innovation on productivity growth across 108 cities in the YREB, underscoring the role of innovation in regional development. Tian et al. [31] conducted a comprehensive study on green innovation, including spatial analysis, coupling coordination, and efficiency evaluation, which is crucial for sustainable development in the region. Wang et al. [32] employed random forest analysis to identify factors affecting urban carbon emissions within the YREB, emphasizing the need for targeted policies to mitigate environmental impacts.

This study can see that there are still some relatively weak points in the urban agglomerations in the YREB, which are obstacles to sustainable, healthy, coordinated development and efficient population expansion in the future. In 2018, the 19th National Congress of the Communist Party of China proposed building a coordinated urban pattern for all sizes and types of cities with urban agglomeration as the main body. Thus far, urban agglomeration—as a special form of regional development and the main carrier of urbanization in China—has attracted the attention of the whole society. In this context, the YREB, as an important strategic region for China's development, is bound to act as a driver of regional development. This study considers the problem of the efficient population size of urban agglomerations in the YREB from a financial perspective. This study find that the weak points of sustainable, healthy, coordinated development in the Yangtze River Delta, Yangtze River Middle Reaches, and Chengdu-Chongqing urban agglomerations are mainly cities with populations of more than 10 million, cities with populations of 3–5 and 5–10 million, and cities with populations of 5–10 and 1–5 million, respectively. If the efficiency of these cities can be further improved, the population sizes can be further expanded.

Based on our findings, to overcome these weaknesses, this study need to broaden fiscal revenues and reduce fiscal expenditures. The path to improving financial efficiency should begin with strategies and countermeasures such as industrial upgrading, structural grading, urban functionalization, transportation networking, and urban-rural integration; give full play to the leading role of cities with populations of more than 10 million in the development of urban agglomerations; and coordinate functional cooperation and development among cities of various population sizes within the urban agglomerations to stimulate the advantages of cities of all population sizes, promote strengths, and avoid weaknesses. In this way, this study can further promote the sustainable, healthy, coordinated development of urban agglomerations in the YREB and enhance their efficient population-carrying capacity.

From a theoretical standpoint, our findings contribute to the understanding of how population size and fiscal efficiency interact within urban agglomerations. By quantifying the optimal population sizes for fiscal efficiency, we provide a new framework for analyzing the sustainability and development potential of urban agglomerations. This framework can be applied to other urban agglomerations, enhancing the generalizability of the research.

For policymakers and urban planners, our results offer practical guidance on managing urban growth and optimizing fiscal resources. The identification of specific population thresholds can inform strategies for regional development, such as targeted investments in

infrastructure and services, and the promotion of balanced growth across cities of different sizes. For example, cities with populations over 10 million in the Yangtze River Delta may benefit from strategies aimed at improving fiscal efficiency through diversification of economic activities and strengthening intra-urban connectivity.

To improve financial efficiency and expand the efficient population sizes of the Yangtze River Delta, Yangtze River Middle Reaches, and Chengdu-Chongqing urban agglomerations, we recommend the following policy measures:

1. Industrial Upgrading: Encourage the transition from traditional industries to high-tech and service-oriented sectors to enhance fiscal revenues.

2. Structural Grading: Develop a hierarchical structure of cities within each agglomeration to distribute functions more evenly and reduce the burden on larger cities.

3. Urban Functionalization: Specialize cities based on their strengths and resources, promoting diverse economic activities.

4. Transportation Networking: Improve intercity transportation networks to facilitate the movement of goods, services, and people.

5. Urban-Rural Integration: Promote balanced development between urban and rural areas to reduce disparities and foster a harmonious regional development environment.

These strategies can help overcome the identified weaknesses and promote sustainable, healthy, and coordinated development in the Yangtze River Economic Belt.

In conclusion, this study provides a novel perspective on the efficient population sizes of urban agglomerations by integrating fiscal efficiency with demographic and urban planning considerations. Our findings have significant implications for both academic research and policy implementation, offering a roadmap for sustainable urban development in the Yangtze River Economic Belt and beyond.

## 5. Conclusion

This study investigates the efficient population sizes of key urban agglomerations in the Yangtze River Economic Belt (YREB) from a fiscal perspective. By analyzing the relationship between population size and fiscal efficiency, we identify the maximum financially efficient population sizes for the Yangtze River Delta, Yangtze River Middle Reaches, and Chengdu-Chongqing urban agglomerations. Our findings reveal that the optimal population sizes for fiscal efficiency vary across these regions, with specific thresholds for different city sizes acting as critical factors.

The core results of this study highlight the importance of balancing population size and fiscal efficiency in urban agglomerations. For the Yangtze River Delta, cities with populations over 10 million are a focal point for improvement. In the Yangtze River Middle Reaches, cities with populations of 3–5 million and 5–10 million need targeted strategies. Similarly, for the Chengdu-Chongqing urban agglomeration, cities with populations of 5–10 million and 1–5 million require specific attention. These insights are crucial for policymakers aiming to optimize urban development and resource allocation.

Our research contributes to the field by providing a quantitative framework for analyzing the efficient population sizes of urban agglomerations. This framework complements existing theories and empirical studies by incorporating a fiscal perspective. Moreover, the practical implications of our findings offer guidance for policymakers and urban planners on how to manage urban growth and optimize fiscal resources effectively.

In summary, this study advances our understanding of the complex interactions between population size and fiscal efficiency in urban agglomerations. It underscores the need for targeted policies that consider both the size and fiscal health of cities within these regions. By doing so, it provides a roadmap for fostering sustainable, healthy, and coordinated development in the YREB and other similar urban agglomerations.

This study not only delves into the issue of effective population size in the Yangtze River Economic Belt urban agglomeration, but also provides a new perspective for understanding the relationship between urban development and population size from a fiscal perspective. For research in other regions, this article provides valuable insights in multiple aspects and can further refine the results to guide practical applications.

## 5.1. Theoretical contributions and inspirations

This article combines knowledge from multiple disciplines such as finance, urban planning, and demography, providing a comprehensive perspective for understanding the development of urban agglomerations. This inspires researchers in other regions to cross the boundaries of traditional disciplines and comprehensively utilize multidisciplinary knowledge and methods when conducting similar research, in order to obtain more comprehensive and in-depth analysis results. From a fiscal perspective, the impact of population size on fiscal revenue and expenditure, public service provision, and other aspects was analyzed. This reminds researchers in other regions that when exploring population size issues, in addition to considering conventional factors such as economy and society, they should also pay attention to the influence of fiscal factors to comprehensively evaluate the rationality of population size.

## 5.2. Policy implications and application values

The practical application areas based on the research results of this article are: (1) Local governments can formulate more precise population policies. For example, in areas with high population inflows, the government can increase investment in public services and improve urban carrying capacity. In areas with high population outflow, the structure of fiscal expenditure can be optimized and the efficiency of fiscal fund utilization can be improved. (2) The urban planning department can draw on the ideas of this study and develop reasonable urban planning plans based on local conditions. By scientifically predicting population size, rational layout of urban space and infrastructure, ensure that urban development is coordinated with population size. (3) According to the trend of population size changes, the government can adjust its public service provision strategy. For example, in the fields of public services such as education, healthcare, and transportation, resources should be allocated reasonably based on changes in population demand to improve the quality and efficiency of public services. (4) Establish a long-term population size monitoring mechanism, regularly collect and analyze relevant data, and timely grasp the trend of population size changes. By comparing and analyzing data from different regions and time periods, provide scientific basis for policy formulation and planning. (5) Establish a population size evaluation system and regularly evaluate the effectiveness of policies. By evaluating the results, adjust policy direction and measures in a timely manner to ensure the achievement of policy objectives.

## 5.3. Research outlook

(1) Focus on improving the methods of data collection and processing to enhance the accuracy and completeness of data. At the same time, unified data collection and statistical standards should be established to better compare and analyze data from different cities and regions. (2) Existing models and methods should be further improved and expanded to comprehensively

consider various factors that affect the effective population size. Especially, the analysis of complex factor interactions should be strengthened to provide more accurate and in-depth insights. (3) In addition to the fiscal perspective, future research should also be comprehensively analyzed from multiple perspectives such as society, economy, and environment. This will help to comprehensively evaluate the development status of urban agglomerations and provide a more comprehensive and scientific basis for policy formulation and planning. (4) More attention should be paid to the role of regional and cultural differences in population size and its impact. By conducting in-depth research on population size changes and their causes in different regions and cultural backgrounds, more specific and targeted recommendations can be provided for the development of urban agglomerations.

## Acknowledgments

This study thanks LetPub (www.letpub.com) for its linguistic assistance during the preparation of this manuscript.

## Author Contributions

**Conceptualization:** Long Xiao.

**Data curation:** Long Xiao.

**Formal analysis:** Long Xiao.

**Investigation:** Long Xiao.

**Methodology:** Long Xiao.

**Writing – original draft:** Long Xiao.

**Writing – review & editing:** Long Xiao.

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
