## [Decision Letter · Decision Letter 0]

30 Apr 2024

PONE-D-24-06627Efficient Population Size of Urban Agglomerations in the Yangtze River Economic Belt from a Financial PerspectivePLOS ONE

Dear Dr. Xiao,

Thank you for submitting your manuscript to PLOS ONE. After careful consideration, we feel that it has merit but does not fully meet PLOS ONE’s publication criteria as it currently stands. Therefore, we invite you to submit a revised version of the manuscript that addresses the points raised during the review process.

We look forward to receiving your revised manuscript.

Kind regards,

Baojie He, Ph.D

Academic Editor

PLOS ONE

Journal Requirements:

"This research was supported by the Major Project of the National social Science Foundation of China (20&ZD066)."

Reviewers' comments:

Reviewer's Responses to Questions

**Comments to the Author**

1. Is the manuscript technically sound, and do the data support the conclusions?

Reviewer #1: Yes

Reviewer #2: Yes

2. Has the statistical analysis been performed appropriately and rigorously? 

Reviewer #1: Yes

Reviewer #2: Yes

3. Have the authors made all data underlying the findings in their manuscript fully available?

Reviewer #1: Yes

Reviewer #2: Yes

4. Is the manuscript presented in an intelligible fashion and written in standard English?

Reviewer #1: Yes

Reviewer #2: Yes

5. Review Comments to the Author

Reviewer #1: The paper can be accepted after making the following minor revisions.

(1) The innovation of this paper needs to be highlighted in the abstract.

(2) The article lacks an important discussion link, in which the author should focus on describing the differences between the article study and other scholars' studies, thus highlighting the relevance and academic value of the article, the following literature should be helpful for your research: (1)Reduction pathways identification of Agricultural Water Pollution in Hubei Province, China. (2) A differential game of water pollution management in the trans-jurisdictional river basin. (3) Coordination of the Industrial-Ecological Economy in the Yangtze River Economic Belt, China.

(3) The article was not written following the correct journal's guidelines to be considered for publication. INTORDCUTION→MRTHOD→RESULTS→DISSCUSION→CONCLUSION

(4) How can your study inspire studies on other regions? The results should be further elaborated to show how they could be used for the real applications.

(5) What are the managerial insights/policy implications of this study? Compared with available literature, what are the theoretical contributions and application values of this study? It is suggested to enhance the corresponding discussions in the conclusion part.

Reviewer #2: The study quantified the effective population size of three urban agglomerations from a financial perspective. By reading the manuscript, I have the following comments and questions:

1.Line 57 of the manuscript fails to provide a detailed elaboration of the universal laws mentioned therein. Furthermore, the text refers to "Western researchers," yet the subsequent cases presented are all Chinese scholars and studies.

2.The introductory section of the manuscript provides a comprehensive overview of existing studies, yet lacks the requisite summaries and generalisations. Furthermore, the necessity, importance, innovativeness, and significance of the study are not adequately addressed in the introductory section.

3.It is recommended that the reasons for the selection of the study area be included in the manuscript. Furthermore, it is advised that the authors provide an overview map of the study area, indicating the location of the cities and their spatial extent.

4.Line 176-Line 183: This section of the statement appears superfluous and lacks any meaningful connection to the study.

5.The figures and their accompanying labels in the manuscript are too small to facilitate the acquisition of the requisite information. Furthermore, the names of the urban agglomerations are incorrectly labelled. It is recommended that the authors incorporate the contents of Figures 2, 3, and 4 into Figure 1 in order to enhance the visual presentation of the results.It is recommended that the authors provide the goodness of fit in Figure 5.

6.The Conclusions and Discussion section, which is largely identical to the results, should be summarized as part of the conclusions. Furthermore, the extended discussion and analysis of the results in the existing manuscript is inadequate to highlight the value of the study.

7.Furthermore, the outcomes of linear/non-linear fitting based on a single factor are frequently characterised by uncertainty and ambiguity. It is therefore recommended that authors consider a multifactor combined analysis in order to enhance the accuracy and representativeness of the results.

In summary, it is recommended that the manuscript be given a major revision.

6. PLOS authors have the option to publish the peer review history of their article (what does this mean?). If published, this will include your full peer review and any attached files.

Reviewer #1: No

Reviewer #2: No

---

## [Author Response · Author response to Decision Letter 0]

16 Jun 2024

Please see attachment file "Response to review comments".

---

## [Decision Letter · Decision Letter 1]

28 Jun 2024

PONE-D-24-06627R1Efficient Population Size of Urban Agglomerations in the Yangtze River Economic Belt from a Financial PerspectivePLOS ONE

Dear Dr. Xiao,

Thank you for submitting your manuscript to PLOS ONE. After careful consideration, we feel that it has merit but does not fully meet PLOS ONE’s publication criteria as it currently stands. Therefore, we invite you to submit a revised version of the manuscript that addresses the points raised during the review process.

We look forward to receiving your revised manuscript.

Kind regards,

Baojie He, Ph.D

Academic Editor

PLOS ONE

Journal Requirements:

Reviewers' comments:

Reviewer's Responses to Questions

**Comments to the Author**

1. If the authors have adequately addressed your comments raised in a previous round of review and you feel that this manuscript is now acceptable for publication, you may indicate that here to bypass the “Comments to the Author” section, enter your conflict of interest statement in the “Confidential to Editor” section, and submit your "Accept" recommendation.

Reviewer #2: All comments have been addressed

2. Is the manuscript technically sound, and do the data support the conclusions?

Reviewer #2: Yes

3. Has the statistical analysis been performed appropriately and rigorously? 

Reviewer #2: Yes

4. Have the authors made all data underlying the findings in their manuscript fully available?

Reviewer #2: Yes

5. Is the manuscript presented in an intelligible fashion and written in standard English?

Reviewer #2: Yes

6. Review Comments to the Author

Reviewer #2: The author has responded to and revised the comments in question in a satisfactory manner, thereby enhancing the quality of the article. However, one minor suggestion remains. With regard to the initial question, the authors have misinterpreted it. It is not a summary and generalisation of this study (Line 118-Line 147), but rather of previous related studies. The authors should therefore revise accordingly. Following further revision, the manuscript can be accepted.

7. PLOS authors have the option to publish the peer review history of their article (what does this mean?). If published, this will include your full peer review and any attached files.

Reviewer #2: No

---

## [Author Response · Author response to Decision Letter 1]

18 Jul 2024

I have upload “Response to Reviewers” file.

---

## [Decision Letter · Decision Letter 2]

25 Jul 2024

PONE-D-24-06627R2Efficient Population Size of Urban Agglomerations in the Yangtze River Economic Belt from a Financial PerspectivePLOS ONE

Dear Dr. Xiao,

Thank you for submitting your manuscript to PLOS ONE. After careful consideration, we feel that it has merit but does not fully meet PLOS ONE’s publication criteria as it currently stands. Therefore, we invite you to submit a revised version of the manuscript that addresses the points raised during the review process.

We look forward to receiving your revised manuscript.

Kind regards,

Xu Xin, Ph.D.

Academic Editor

PLOS ONE

Journal Requirements:

Additional Editor Comments:

Dear Dr. Xiao,

Greetings!

Thank you very much for your submission to PLOS ONE.

I am pleased to inform you that Reviewer 1 recommended a "minor revision" during the first round of revisions. Therefore, the academic editor (AE) did not invite him to the subsequent review. Reviewer 2 has already recommended a "accept" at the second round of revision (R2). I agree with their comments and think that this manuscript is basically ready for acceptance. However, after my careful assessment, I still have some concerns about this manuscript. I think it is necessary for the author to emphasize the following before this manuscript is accepted.

Comments:

1. The author's correspondence email address should be an institutional email address. Please try to use the university official email instead of Outlook email.

2. Abstract is very boring. Please hit the pillars of academic abstract (i.e., background information, methods, important results, implications in brief). The current abstract lacks background information, the methods are not clearly presented, and the results and significance are too lengthy.

3. Introduction: Please don't start by arguing about China. This is an international journal. Please start by giving some background information suitable for international readers. The format of "Wang et al. [5] - [8]" is incorrect.

4. The introduction section should end with a paragraph describing the organization of the subsequent sections. The research processes for this manuscript should be given (see Figure 1 of the reference given below).

5. Some sentences are so strange that the authors need to find a professional agency to proofread them. For example, Line 187, "The data required for this study involve the populations of urban agglomerations, fiscal revenues, and fiscal expenditures, among other data.", What do you mean by among other data? The figure of the study area needs to be redrawn, and the source and type of data need to be organized in tables (see Table 1 in reference attached below).

6. Line 208: Based on Nakamura and Kanauchi’s [12], This way of writing is wrong.

7. In equations 1 and 2, all parameters/variables (i.e., N, Y, C and E) need to be italicized. Meanwhile, I don't understand the meaning of functions f1 and f2. Line 215: "By deriving N in Equation (2), the maximum value of Y/C can be calculated." I find it hard to understand what this sentence means.

8. Line 254: R2 should be deleted.

9. Line 301: There is a problem with the format of the formula. All equations in the manuscript should be edited using software such as Mathtype.

10. Discussion section should be improved. It should present core results that emerge from the study, interprets and discusses them (by the authors). The discussion section needs to compare this study with other key studies, emphasize the significance of this study, provide implication for academia (theoretical) and policy and decision makers (managerial) (what benefit should be gained out of such result, what does it mean to port managers and policy makers). If I read the study, what is the benefit I gain as new thing? Readers, researchers and policy makers, should leave with a bunch relevant implication not only results. Some of the management insights given so far are so obvious that they can be obtained even without conducting this research. For the specific writing method of this section, the author can refer to the reference attached below.

11. The conclusion should be organized better. That is, provide a summary of the study, brief results, in addition to proper limitations and future research areas. The conclusion currently lacks the simple introduction and specific conclusions of this article. For the specific writing method of this section, the author can refer to the reference attached below.

12. The formatting of the references in this manuscript needs revision. I often see inappropriate citations. The autorh should also cite some articles in this journal.

13. The number of cited literature is relatively small (only 27 literature), which is also related to the insufficient literature review. I suggest that the author strengthen his/her review of existing literature in the introduction section.

In addition to the problems I pointed out, the author also needs to carefully sort out the flow of the manuscript and try to make the manuscript clear and fluent. At the same time, I think the format of the following article is more reasonable for the author's reference.

Reference:

Zhang, T., Xin, X., He, F., Wang, X., & Chen, K. (2023). How to promote sustainable land use in Hangzhou Bay, China? A decision framework based on fuzzy multiobjective optimization and spatial simulation. Journal of Cleaner Production, 414, 137576.

Reviewers' comments:

Reviewer's Responses to Questions

**Comments to the Author**

1. If the authors have adequately addressed your comments raised in a previous round of review and you feel that this manuscript is now acceptable for publication, you may indicate that here to bypass the “Comments to the Author” section, enter your conflict of interest statement in the “Confidential to Editor” section, and submit your "Accept" recommendation.

Reviewer #2: All comments have been addressed

2. Is the manuscript technically sound, and do the data support the conclusions?

Reviewer #2: Yes

3. Has the statistical analysis been performed appropriately and rigorously? 

Reviewer #2: Yes

4. Have the authors made all data underlying the findings in their manuscript fully available?

Reviewer #2: Yes

5. Is the manuscript presented in an intelligible fashion and written in standard English?

Reviewer #2: Yes

6. Review Comments to the Author

Reviewer #2: (No Response)

7. PLOS authors have the option to publish the peer review history of their article (what does this mean?). If published, this will include your full peer review and any attached files.

Reviewer #2: No

---

## [Author Response · Author response to Decision Letter 2]

10 Sep 2024

I have upload Response to Reviewer's Comments in your system.

---

## [Editor Report · Decision Letter 3]

13 Sep 2024

Efficient Population Size of Urban Agglomerations in the Yangtze River Economic Belt from a Financial Perspective

PONE-D-24-06627R3

Dear Dr. Xiao,

We’re pleased to inform you that your manuscript has been judged scientifically suitable for publication and will be formally accepted for publication once it meets all outstanding technical requirements.

Kind regards,

Xu Xin

Academic Editor

PLOS ONE

---

## [Editor Report · Acceptance letter]

19 Sep 2024

PONE-D-24-06627R3 

PLOS ONE

Dear Dr. Xiao, 

I'm pleased to inform you that your manuscript has been deemed suitable for publication in PLOS ONE. Congratulations! Your manuscript is now being handed over to our production team.

Kind regards, 

on behalf of

Dr. Xu Xin 

Academic Editor

PLOS ONE